# Efficacy of acupuncture on lower limb motor dysfunction following stroke: A systematic review and meta-analysis of randomized controlled trials

Peng Chen[1]☯, Xing Jin[2,3,4]☯, Debiao Yu[2,3,4], Xiaoting Chen[5], Yaoyu Lin[5], Fuchun Wu[2,3,4], Bin Shao ᴵᴰ [1,2,3,4]*

1 College of Acupuncture and Tuina, Fujian University of Traditional Chinese Medicine, Fuzhou, Fujian, China, 2 Rehabilitation Medicine Center, Fujian Provincial Hospital, Fuzhou, Fujian, China, 3 Provincial Clinical Medical College of Fujian Medical University, Fuzhou, Fujian, China, 4 Rehabilitation Medicine Center, Fuzhou University Affiliated Provincial Hospital, Fuzhou, Fujian, China, 5 College of Rehabilitation Medicine, Fujian University of Traditional Chinese Medicine, Fuzhou, Fujian, China

☯ These authors contributed equally to this work.
* slkfsb@sina.com

## Abstract

### Background

Acupuncture is widely used for Lower Limb Motor Dysfunction Following Stroke (LLMD) in China, though its effectiveness remains unclear. This meta-analysis aims to evaluate the effectiveness of acupuncture for LLMD.

### Methods

We searched eight databases, including PubMed, Embase, Web of Science, Cochrane Library, China National Knowledge Infrastructure, Wanfang, VIP, and CBM, up to December 2023. Randomized controlled trials on acupuncture therapy for LLMD after stroke were included in this study. Outcome measures included motor function, balance function, walking ability, and daily living activities. Two researchers conducted independent literature screening, data extraction, and quality assessment in accordance with Cochrane Collaboration network's standards. Review Manager 5.3 and Stata 17.0 were used in data analysis. Results were presented as mean difference (MD) or standardized mean difference (SMD) with 95% confidence interval (95% CI).

### Results

Twelve studies involving 1318 patients, most of which showed low or unclear risk of bias, were included in this review. Meta-analysis results indicate that compared with conventional treatment, acupuncture intervention can improve scores in Fugl–Meyer

**Data availability statement:** All relevant data are within the manuscript and its Supporting Information files.

**Funding:** Fund Project of Fujian Provincial Natural Science Foundation in 2022 (No. 2022J011019) and the 2022 Fujian Provincial Health and Health Youth Backbone Training Project (No. 2022GA009). The funders had no role in study design, data collection and analysis, decision to publish, or preparation of the manuscript.

**Competing interests:** The authors have declared that no competing interests exist.

**Abbreviations:** LLMD, lower limb motor dysfunction; 95% CI, 95% confidence interval; FMA-L, Fugl–Meyer Assessment for Lower; BBS, Berg Balance Scale; FAC, functional ambulation category scale; MBI, Modified Barthel Index; OR, odds ratio; RCT, randomized controlled trial; REM, random-effects model; SMD, standardized mean difference.

Assessment for lower scale (SMD = -0.48, 95% CI [-0.92, –0.04], Z = 2.16, $P$ = .03), Berg Balance Scale (SMD = -0.86, 95%CI [-1.65, –0.07], Z = 2.14, $P$ = .03), Functional Ambulation Category scale (SMD = -0.74, 95%CI [-2.33, 0.84], Z = 0.92, $P$ = .36), and Modified Barthel Index Scale (SMD = 0.27, 95%CI [-0.30, 0.84], Z = 0.94, $P$ = .35).

## Conclusion

The results of this study suggest that acupuncture combined rehabilitation training may be more effective than conventional rehabilitation alone in improving LLMD, balance function, walking ability, and daily living activities after stroke. Despite limitations due to the low quality of the included studies and methodological constraints, acupuncture combined with rehabilitation training may serve as an effective approach for the treatment of LLMD poststroke.

---

## 1. Introduction

Ischemic stroke, characterized by the abrupt loss of cerebral blood flow due to an occlusion, precipitates a cascade of pathophysiological events culminating in extensive neuronal damage [1]. Approximately 12.2 million new cases of stroke were recorded each year, with over 80 million stroke survivors globally [2]. Moreover, the incidence of stroke, which severely impacts individual health, has been increasingly affecting younger populations [3]. Hemiparesis is a typical sequela among survivors, with over 50% experiencing substantial lower limb motor dysfunction (LLMD) [4]. This condition significantly compromises a patient's ability to walk and maintain postural stability, thereby affecting their quality of daily life [5]. Impaired lower limb motor function also hampers the rehabilitation process, diminishes individual autonomy, and restricts the patients' ability to engage in social roles, which further exacerbates their physical and psychological burdens [6,7]. Various treatments, including pharmacotherapy, physical therapy, rehabilitation training, assistive devices, and supportive equipment, have been developed to address poststroke motor deficits. However, these interventions pose different side effects and may decrease patients' and confidence of patients due to prolonged rehabilitation training [8–10].

Acupuncture, originating in China, is a traditional treatment method based on the theory of meridians in traditional Chinese medicine[11]. Acupuncture therapy has been widely used in the treatment of movement disorders secondary to diseases, such as stroke, brainstem injury, Parkinson's disease, spinal cord injury, multiple sclerosis, and peripheral nerve and muscle injuries [12,13]. Studies have demonstrated that acupuncture can improve motor function in various conditions, including poststroke motor dysfunction, through regulates nervous system function, promotes blood circulation, and relieves muscle spasms with few side effects and risks [14,15]. Recent studies have shown that acupuncture promotes the expression of neurotrophic factors, enhances neuroplasticity and promotes neuronal dendritic remodelling and synapse formation [12]. And other studies have confirmed that acupuncture can improve the functional connectivity between the motor cortex and

cerebellar regions, optimising the integration of the brain's motor networks [16]. Studies have confirmed that acupuncture increases local blood flow and improves microcirculation and tissue oxygenation by modulating the expression of vasodilatory factors, as well as acupuncture's ability to modulate motor neuron excitability, helping to restore movement patterns through muscle synergy [17].

Clinical studies have demonstrated the capability of acupuncture to improve walking ability and lower-limb balance, strengthen lower-limb muscles, and enhance the gait of stroke patients [18,19]. However, other research suggests a non-significant recovery effect of acupuncture on poststroke LLMD [20]. Given these conflicting findings, further investigation is warranted. Therefore, this meta-analysis was conducted to evaluate the therapeutic effects of acupuncture on LLMD after stroke by assessing its impact on motor function, balance function, walking ability, and daily living activities, aiming to provide evidence-based support for clinical applications of acupuncture in poststroke LLMD treatment.

## 2. Materials and methods

This systematic review and meta-analysis involves analyzing data from previously published research rather than direct engagement with patients or access to personal health information. Therefore, according to international guidelines for research ethics, this type of investigation does not require fresh ethical approval or patient consent.

### 2.1. Study registration

This protocol has been registered in the International Prospective Register of Systematic Reviews (PROSPERO) under trial registration number CRD42023487617.

### 2.2. Search strategy

The search strategy was implemented in accordance with the Cochrane Handbook guidelines (5.1.0). We searched eight electronic databases from their inception to December 31, 2023:Embase, PubMed, Cochrane Library, Web of Science, China National Knowledge Infrastructure, WanFang, China Science Journal Database (VIP), and Chinese Biomedical Literature Database. Additionally, we searched for ongoing in the WHO International Clinical Trials Registry Platform and the Chinese Clinical Trial Registry. The corresponding authors were contacted in case of incomplete data. The search strategy was tailored to each database using appropriate combinations of subject headings and free terms. The detailed search strategy for PubMed is provided as an example in Table 1.

[Insert keywords for other databases (Appendix 1)].

### 2.3. Inclusion criteria

Study Type: Randomized controlled trials (RCTs) investigating the effects of acupuncture on poststroke lower limb motor function and limited to publications in Chinese and English.

Study Participants: Patients aged 18–80 years with stroke confirmed by cranial computed tomography or magnetic resonance imaging, exhibiting LLMD, maintaining clear consciousness, and capable of cooperation during examinations and treatment.

Interventions: The control group received routine training (rehabilitation, sensory motor, drug therapy + rehabilitation, rehabilitation, and functional training); the observation group underwent acupuncture treatment.

Outcome Measures: Motor function, balance function, walking ability, and daily living activities assessed through the Fugl–Meyer Assessment for Lower Scale (FMA-L) [21], Berg Balance Scale (BBS) [22], Functional Ambulation Category scale (FAC), and Modified Barthel Index scale (MBI), respectively.

The selection of assessment tools (FMA-L, BBS, MBI) was based on their established validity and widespread clinical application in stroke rehabilitation research. The FMA-L has been validated as a comprehensive tool for evaluating lower extremity motor function in stroke patients. The BBS is widely recognized as a reliable measure for assessing balance

**Table 1. detailed search strategy for PubMed.**

| Process | Accession number |
| --- | --- |
| | **PubMed search strategy.** |
| #1 | "Cerebrovascular Disorders"[Mesh] OR "Basal Ganglia Cerebrovascular Disease"[Mesh] OR "Brain Ischemia"[Mesh] OR "Carotid Artery Diseases"[Mesh] OR "Cerebral Small Vessel Diseases"[Mesh] OR "Intracranial Arterial Diseases"[Mesh] OR "Intracranial Embolism and Thrombosis"[Mesh] OR "Intracranial Hemorrhages"[Mesh] OR "Stroke"[Mesh] OR "Brain Infarction"[Mesh] OR "Stroke, Lacunar"[Mesh] OR "Vasospasm, Intracranial"[Mesh] OR "Vertebral Artery Dissection"[Mesh] |
| #2 | stroke [Title/Abstract] OR "post-stroke" [Title/Abstract] OR apoplexy [Title/Abstract] OR "cerebral vascular" [Title/Abstract] OR "brain vascular" [Title/Abstract] OR "cerebrovascular" [Title/Abstract] OR cva [Title/Abstract] OR "subarachnoid hemorrhage" [Title/Abstract] |
| #3 | brain [Title/Abstract] OR cerebrum [Title/Abstract] OR cerebellum [Title/Abstract] OR "vertebrobasilar" [Title/Abstract] OR hemisphere [Title/Abstract] OR intracranial [Title/Abstract] OR intracerebral [Title/Abstract] OR infratentorial [Title/Abstract] OR supratentorial [Title/Abstract] OR "Middle Cerebral Artery" [Title/Abstract] OR "MCA" [Title/Abstract] OR "Anterior Circulation" [Title/Abstract] OR "Posterior Circulation" [Title/Abstract] OR "Basilar Artery" [Title/Abstract] OR "Vertebral Artery" [Title/Abstract] OR "space-occupying" [Title/Abstract] |
| #4 | ischemia [Title/Abstract] OR infarction [Title/Abstract] OR thrombosis [Title/Abstract] OR embolism [Title/Abstract] OR occlusion [Title/Abstract] OR hypoxia [Title/Abstract] |
| #5 | brain [Title/Abstract] OR cerebrum [Title/Abstract] OR cerebellum [Title/Abstract] OR intracerebral [Title/Abstract] OR intracranial [Title/Abstract] OR parenchymal [Title/Abstract] OR intraparenchymal [Title/Abstract] OR intraventricular [Title/Abstract] OR infratentorial [Title/Abstract] OR supratentorial [Title/Abstract] OR "basal ganglia" [Title/Abstract] OR "putaminal" [Title/Abstract] OR putamen [Title/Abstract] OR "Posterior Fossa" [Title/Abstract] OR hemisphere [Title/Abstract] OR subarachnoid [Title/Abstract] |
| #6 | #1 OR #2 OR #3 OR #4 OR #5 |
| #7 | "Motor Disorders"[Mesh] OR "Hemiplegia"[Mesh] OR "Paresis"[Mesh] OR "Gait Disorders, Neurologic"[Mesh] OR "Motor Skills Disorders"[Mesh] OR "Muscle Weakness"[Mesh] OR "Mobility Limitation"[Mesh]) AND ("Lower Extremity"[Mesh] OR "Leg"[Mesh] OR "Foot"[Mesh] OR "Walking"[Mesh] |
| #8 | "lower limb" [Title/Abstract] OR motor disorders [Title/Abstract] OR hemiplegia [Title/Abstract] OR hemiparesis [Title/Abstract] OR paresis [Title/Abstract] OR paraplegia [Title/Abstract] OR "motor function" [Title/Abstract] OR "motor skills" [Title/Abstract] OR "muscle strength" [Title/Abstract] OR "muscle weakness" [Title/Abstract] OR "gait disorders" [Title/Abstract] OR "walking difficulties" [Title/Abstract] OR "mobility limitations" [Title/Abstract] OR "lower extremity function" [Title/Abstract] OR "leg function" [Title/Abstract] OR "foot function" [Title/Abstract] OR "lower limb function" [Title/Abstract] |
| #9 | #7 OR #8 |
| #10 | "Acupuncture Therapy"[Mesh] OR acupuncture [Title/Abstract] OR acupoint* [Title/Abstract] OR "electroacupuncture" [Title/Abstract] OR "auriculotherapy" [Title/Abstract] OR "scalp acupuncture" [Title/Abstract] OR "laser acupuncture" [Title/Abstract] OR "manual acupuncture" [Title/Abstract] OR "acupuncture analgesia" [Title/Abstract] OR "acupressure" [Title/Abstract] |
| #11 | "Randomized Controlled Trials as Topic"[Mesh] OR "Random Allocation"[Mesh] OR "Controlled Clinical Trials as Topic"[Mesh] OR "Control Groups"[Mesh] OR "Clinical Trials as Topic"[Mesh] OR "Clinical Trials, Phase I as Topic"[Mesh] OR "Clinical Trials, Phase II as Topic"[Mesh] OR "Clinical Trials, Phase III as Topic"[Mesh] OR "Clinical Trials, Phase IV as Topic"[Mesh] OR "Double-Blind Method"[Mesh] OR "Single-Blind Method"[Mesh] OR "Placebos"[Mesh] OR "Placebo Effect"[Mesh] OR "Cross-Over Studies"[Mesh] OR "Randomized Controlled Trial"[Publication Type] OR "Controlled Clinical Trial"[Publication Type] OR "Clinical Trial"[Publication Type] OR "clinical trial, phase i"[Publication Type] OR "clinical trial, phase ii"[Publication Type] OR "clinical trial, phase iii"[Publication Type] OR "clinical trial, phase iv"[Publication Type] OR random* OR RCT OR RCTs |
| #12 | #6 AND #9 AND #10 AND #11 |
| #13 | #12 Filters: from 1974/1/1–2023/12/31 |

function, while the MBI has been established as a standard tool for evaluating activities of daily living in rehabilitation settings. However, we acknowledge that these assessment tools may have limitations in capturing subtle changes in motor recovery and functional improvements during the rehabilitation process.

## 2.4. Exclusion Criteria

Duplicate publications; non-RCTs including animal experiments, literature reviews, systematic reviews, meta-analyses, case reports, expert summaries, operational research, parameter studies, prospective or retrospective clinical observations, comments, letters, conference abstracts, etc.; studies with incomplete data where authors could not be reached or did not respond; studies involving patients with swallowing difficulties due to conditions other than stroke, such as

brainstem injury, Parkinson's disease, spinal cord injury, multiple sclerosis, peripheral nerve and muscle injuries, etc.; Studies where the intervention group received additional treatments other than acupuncture compared to the control group

### 2.5. Literature selection

Two researchers independently conducted the literature search and screening process. Initially, all literatures retrieved from the preliminary search were imported into EndNote (Clarivate Analytics 21.0), and duplicates were removed using software functions; highly irrelevant literatures were excluded beyond reading titles, abstracts, and keywords. Subsequently, the two researchers independently reviewed the full texts of potentially eligible literature. Those that failed to meet the criteria were deleted, and reasons for study exclusion were recorded. Moreover, a third party was assigned to conduct a reassessment and make a decision when disagreements occurred between the two researchers. The study selection process was described using the Preferred Reporting Items for Systematic Reviews and Meta-Analyses flow diagram.

### 2.6. Data extraction

The reviewers also independently performed data extraction from the studies. After comparison of their results and verification with the original articles, the accuracy and completeness of each data point were confirmed. Extracted data mainly included author, publication year, sample size, patient age, interventions, treatment process, and outcome measures. The corresponding author was contacted for information that needed supplementation.

### 2.7. Assessment of risk of bias (ROB)

The two independent researchers assessed the risk of bias using the methods endorsed by The Cochrane Collaboration, which included the following domains: (a) randomization process; (b) deviations from intended interventions; (c) missing outcome data; (d) measurement of the outcome; (e) selection of the reported result. Any disagreements were resolved by discussion.

### 2.8. Statisticval methods

Statistical analysis was conducted using Review Manager 5.3 software (Cochrane, England). Odds ratio (OR) and 95% confidence interval (CI) were used as combined effect size indicators for categorical variables, and standardized mean difference (SMDs) were applied for continuous variables. The choice of random-effects model was justified by the anticipated clinical heterogeneity among studies due to variations in acupuncture techniques, rehabilitation protocols, and patient characteristics. SMD and 95% CI were used as indicators of the combined effect size. In case of missing data, studies reporting their outcomes were included in the statistical analysis. In case of incompletely reported required change amount, the reviewers consulted the Cochrane Handbook to perform manual calculations of the mean and standard deviation based on the baseline and outcome data reported. $I^2$ and $ChI^2$ tests were conducted to determine the heterogeneity of included studies. For studies with zero heterogeneity or low heterogeneity ($I^2 < 50\%$, $P > 0.1$), a fixed-effects model was used for analysis. In cases of high heterogeneity ($I^2 \geq 50\%$, $P < 0.1$), a random-effects model (REM) was used for the analysis. In cases of significant heterogeneity, efforts were exerted to identify potential sources and consider subgroup or sensitivity analyses. Sensitivity analyses were conducted to assess the robustness of results and identify potential outliers. Descriptive analysis was conducted when data could not be synthesized. Funnel plot analysis and Egger test (performed in Stata 17.0 software) were used to assess the risk of publication bias with respect to outcome measures in $\geq 10$ included studies. If the distribution in the funnel plot was approximately symmetric and the P value of the Egger test was $> 0.05$, it indicated no significant bias and the conclusion was reliable.

## 3. Results

### 3.1 Study selection

The initial search strategy yielded 2,718 records for consideration in this work. The initial screening process identified and excluded 1,341 duplicates, unqualified items flagged by automated tools, and records excluded for other reasons. Among the remaining 1,377 articles, following the inclusion criteria, we excluded guidelines, animal experiments, reviews, data mining studies, operational studies, correlational studies, non-acupuncture intervention studies, and non-RCT papers. Finally, 12 RCTs [23–34] were included. Fig 1 depicts the literature screening process and results.

### 3.2. Study characteristics

This meta-analysis included 12 trials with a total of 1,318 patients. All studies were published between 2011 and 2023, were described as RCTs, and were published in Chinese or English. The studies had sample sizes ranging from 57 to 240, with similar proportions of male and female participants and average age. Treatment lasted from 4 weeks to 8 weeks. Table 2 presents the general information on the included trials.

### 3.3. Risk of bias in included studies

The Cochrane Handbook was used to assess the risk of bias in the included studies. Twelve included studies, with ten trials [23–25,27–32] (83.33%) having a low risk of bias in terms of the generation of random sequence, which was generated using a random number table or computer software. Two trials [26,34] (16.67%) lacked detailed reports on the method used for random sequence generation, and one trials[29] (8.33%) applied allocation concealment. Most trials lacked description for their blinding methods, but four[23,24,26,29] (33.33%) reported blinding of outcome assessment. Assessment of

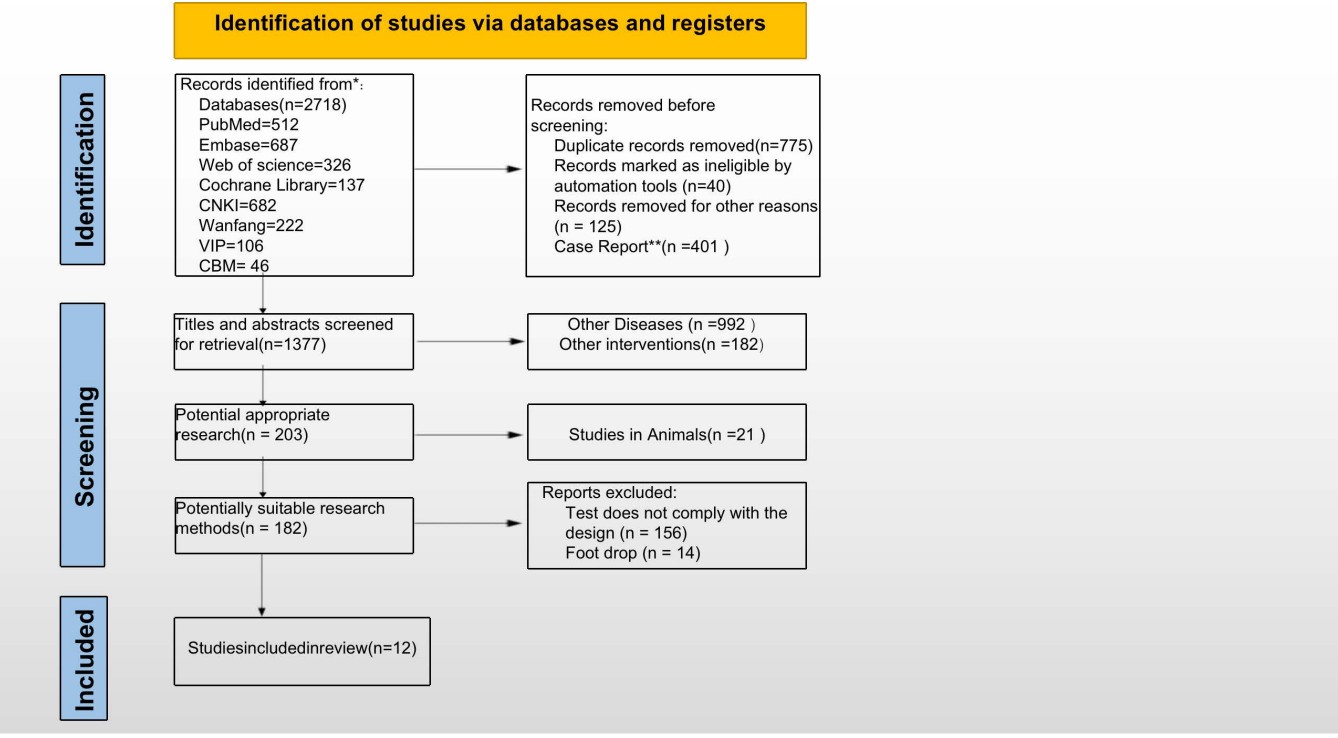

**Fig 1. The flowchart of search results for meta-analysis.**

**Table 2. Basic characteristics of the included studies.**

| Name | time | sample size EG/CG | gender ratio EG/CG | course of disease EG/CG | Age EG/CG | intervention EG/CG | time | outcome measures |
|---|---|---|---|---|---|---|---|---|
| WeijunGong | 2009 | 124/116 | 63:61/58:58 | | 57.8/58.2 | EA/rehabilitation | 6week | FMA-L, CSS, Gait Analysis |
| ZHANG Shao-huag | 2022 | 70/72 | 35:35/40:32 | 12.44±4.4/11.79±4.7 | 57.35±15.49/55.54±17.19 | IDSA/TSA | 4week | FMA-L, BBS, 6MWT, MBI, Gait Analysis |
| Chu Jiamei | 2015 | 48/48 | 29:19/30:18 | 5.89±4.69/5.84±4.03 | 67±11.1/67.1±11.2 | acupuncture/rehabilitation | 4week | FMA-L, BBS, TCT, FAC, 10MWT |
| Xu Lei | 2022 | 31/31 | 15:16/17:14 | 6.01±1.44/5.64±1.51 | 60±7.4/59.3±6.7 | acupuncture/rehabilitation | 4week | FMA-L, BBS, BI, sEMG |
| Wang Yahui | 2016 | 30/30 | 16:14/15:15 | | 66±6/65±7 | acupuncture/acupuncture with rehabilitation | 4week | FMA-L, HA-MA, HAMD, CSS |
| Li Yuqin | 2014 | 45/45 | 26:19/24:21 | 3.67±1.52/3.6±1.43 | 63.64±9.33/64.66±8.52 | abdominal acupuncture/acupuncture | 4week | FMA-L, MAS, CSI, PRO |
| Li Fei | 2019 | 32/32 | 15:17/16:16 | 2.14±0.57/2.29±0.43 | 56±8/57±10 | EA/PNF | 4week | FMA-L, MBI, Proprioception, Clinical Efficacy |
| Wang Xinwei | 2017 | 39/39 | 20:19/21:18 | 3.12±1.48/3.44±1.59 | 56.61±14.36/55.33±15.49 | EA/acupuncture | 4week | FMA-L, MAS, FAC, Tinetti Gait |
| Li Xiupeng | 2019 | 30/30 | 21:9/19:11 | 10.23±3.54/10.56±3.21 | 52.71±7.52/53.35±6.21 | acupuncture therapy /acupuncture | 4week | FMA-L, MAS, CSS, Clinical Efficacy |
| Lai Jinshu | 2023 | 29/28 | 16:13/18:10 | 63 (30,95.5)/66.5 (23,106) | 51.62±9.14/50.57±8.58 | acupuncture therapy acupuncture with rehabilitation | 4week | FMA-L, BBS, sEMG |
| Zheng Shitian | 2020 | 31/31 | 17:14/16:15 | 49 (35,60)/45 (38,63) | 55 (51,58)/51 (42,63) | acupuncture therapy acupuncture with rehabilitation | 4week | FMA-L, BBS, FAC, sEMG |
| Wu Weikun | 2023 | 32/32 | 18:14/17:15 | 49.50 (27.25,74.25)/44.50 (31.00,65.00) | 48.47±5.36/52.00 (46.25,55.00) | EA/ acupuncture with rehabilitation | 4week | FMA-L, BBS, MBI, sEMG |
| Fwng Junwei | 2023 | 32/32 | 14:18/20:12 | 19.50 (17.00,30.75)/24.50 (18.00,40.00) | 61.25±11.64/58.88±12.40 | Reverse the needle /Wake-up call the needle | 4week | FMA-L, MAS, MBI, VAS |

FMA-L = Fugl–Meyer Assessment for Lower, CSS = Composite spasticity scale, BBS = Berg Balance Scale, 6MWT = 6 minute walk test, MBI = Modified Barthel Index, TCT = Trunk Control Test, BI = Barthel Index,

MAS = Modified Ashworth Scale, CSI = Clinic Spasticity Index, PRO = patient reported outcome, FAC = Functional Ambulation Category scale, VAS = Visual Analogue Scale

A

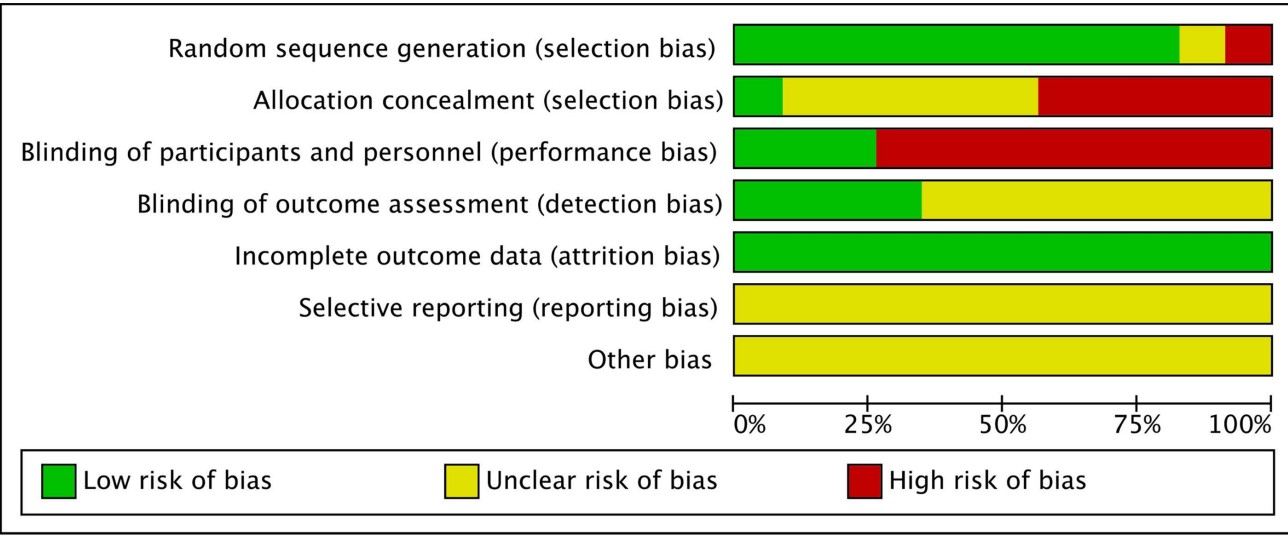

B

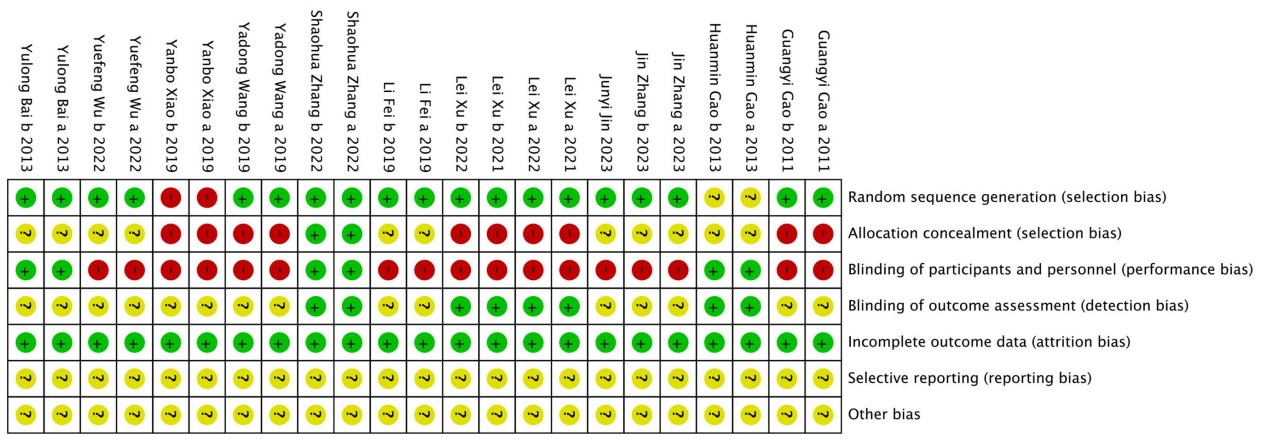

**Fig 2. (A) Risk of bias graph. (B)** Risk of bias summary.

selective reporting of results caused difficulty due to the absence of protocols for the included trials. Based on descriptions in their methodology section, all included trials had a low level of ROB. Thus, the overall quality of the 12 included trials was at a low risk (Figs 2A and 2B).

### 3.4. Results analysis

**3.4.1. FMA-L score.** Eight trials [24,25,29–34] (66.67%) involving 932 participants reported FMA-L as the primary outcome. The findings showed that the experimental group demonstrated significantly greater improvement in FMA-L compared to the control (SMD = −0.48, 95%CI [−0.92,-0.04], I2 = 89%, P < 0.05 random-effects mode)(Fig 3). Significant heterogeneity was observed. As FMA-L included ≥10 studies, so meta-regression and subgroup analysis were conducted to explore the sources of heterogeneity. Meta-regression suggested that the tested variables including treatment duration

A

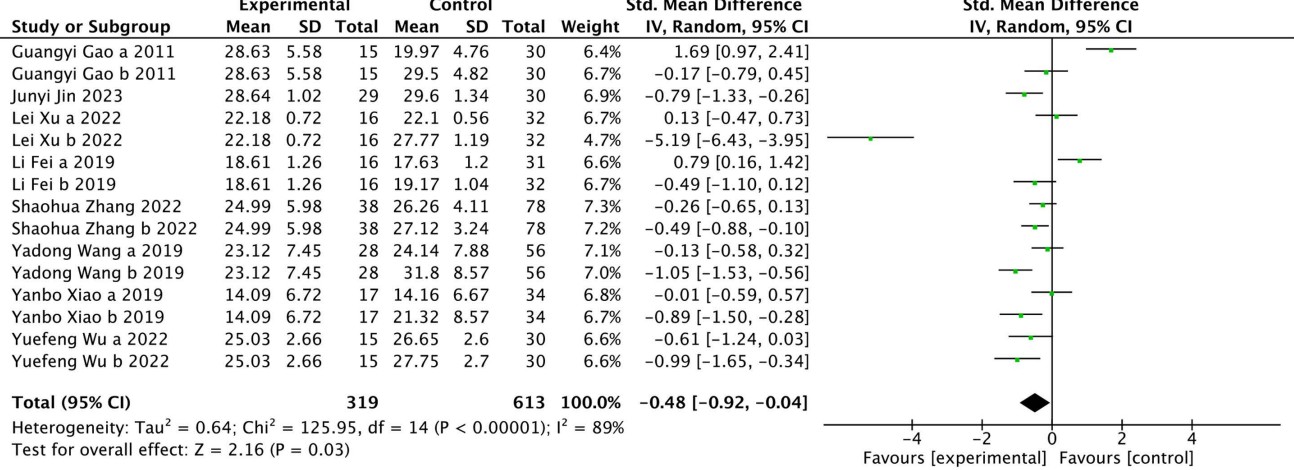

B

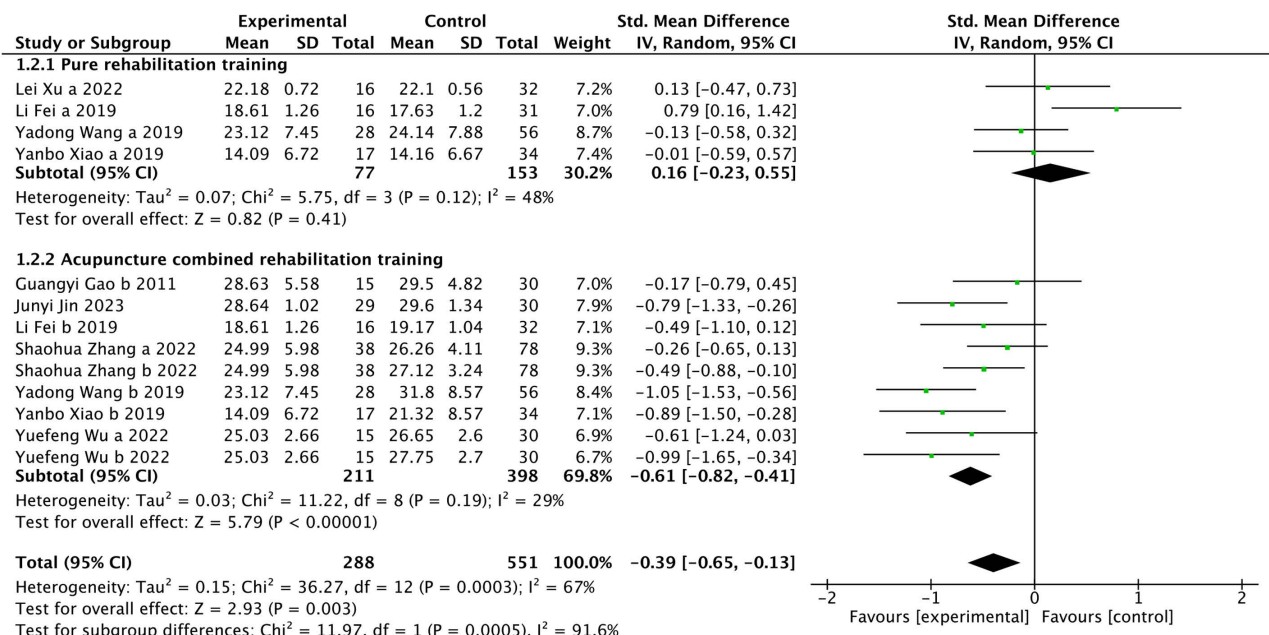

**Fig 3. The forest plot of FMA-L.** (A) Meta-analysis results containing FMA-L for all studies. (B) Results of subgroup analyses of FMA-L.

(P = 0.44), course of disease (P = 0.44), baseline level (P = 0.32) might not be the source of heterogeneity. Subgroup analyses were performed according to the different intervention modalities in the control group. The results showed no significant difference in lower limb motor function improvement between acupuncture and rehabilitation alone (SMD = 0.16, 95% CI [-0.23, 0.55], Z = 0.82, P = .41); However, acupuncture showed better improvement in lower limb motor function compared to combined treatment (SMD = -0.61, 95% CI [-0.82, -0.41], Z = 5.79, P < .001) (Fig 3B). Sensitivity analysis revealed that the results lacked robustness (see Fig 4). When examining publication bias, both the funnel plot (Fig 5A) and Egger's test (P = 0.368 > 0.05, Fig 5B) indicated no evidence of such bias.

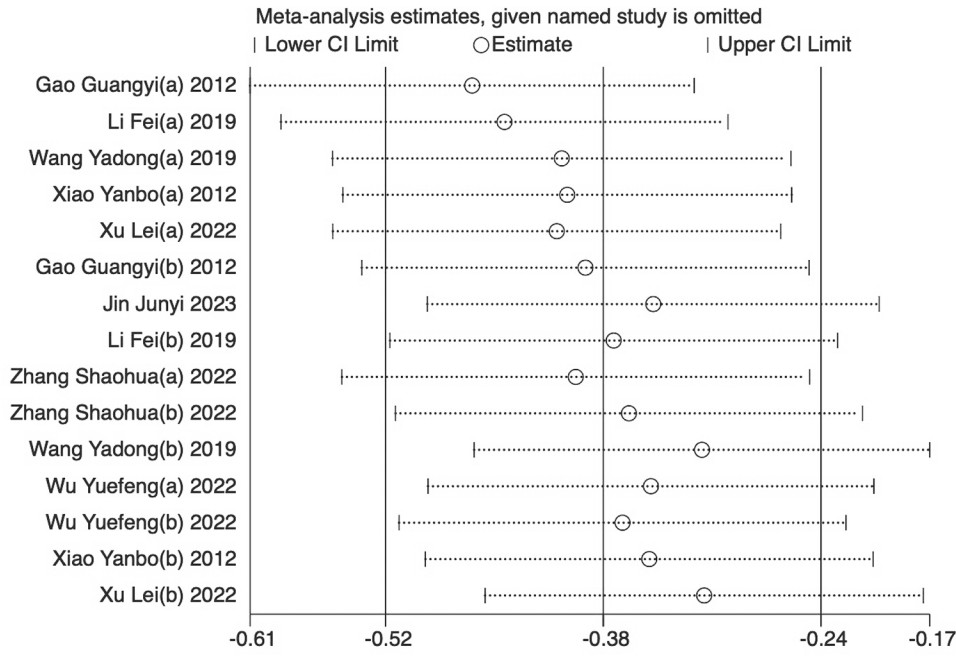

**Fig 4. Sensitivity analysis results for FMA-L.**

**3.4.2. BBS score** Four trials [23,24,28,29] (33.33%) involving 570 participants reported BBS as the primary outcome. Our findings showed that the experimental group demonstrated significantly greater improvement in BBS compared to the control group (SMD = −0.86, 95%CI [-1.56, -0.07], I2 = 94%, P < .05, random-effects mode)(Fig 6A). Significant heterogeneity was observed. BBS included ≥10 studies, so meta-regression and subgroup analysis were used to explore the source of heterogeneity. Meta-regression suggested that disease duration (P > .76) and baseline levels (P > .07) might not be the source of heterogeneity. However, the control group intervention type (P > .03) may be a source of heterogeneity. Subgroup analyses were performed according to different intervention modalities in the control group. The results showed that acupuncture was more effective than rehabilitation alone in improving patients' balance (SMD = 0.61, 95% CI [0.22, 1.00], Z = 3.09, P < 0.05); There was no statistically significant difference between acupuncture and acupuncture combined with lower limb robotic exercise (SMD = -0.20, 95% CI [- 0.47, 0.08], Z = 1.4, P = 0.16); However, acupuncture showed superior effectiveness combined to acupuncture combined with rehabilitation training. (SMD = -3.52, 95% CI [-4.19, -2.85], Z = 10.3, P < .001) (Fig 6B). Sensitivity analysis revealed that the results lacked robustness (Fig 7). When examining publication bias, both the funnel plot (Fig 8A) and Egger's test (P = 0.382 > 0.05, Fig 8B) indicated no evidence of such bias.

**3.4.3. MBI score.** Five trials [25,26,29,31,33] (41.67%) that involved 664 participants reported MBI as the primary outcome. The experimental group showed superior improvement in MBI compared with the control group (SMD = 0.27, 95%CI[-0.30, 0.84], I2 = 90%, *P* = .35, random-effects mode) (Fig 9A). There was significant heterogeneity. As MBI included ≥10 studies, meta-regression and subgroup analysis were used to explore the source of heterogeneity. Meta-regression suggested that the tested variables including treatment duration (*P* > .39), course of disease (*P* > .14) might not be the source of heterogeneity. Subgroup analyses were performed according to the different intervention modalities in the control group. The results showed that acupuncture intervention was not statistically significant compared with rehabilitation alone (SMD = 0.2, 95% CI [-0.41, 0.8], Z = 0.63, *P* = .53), but acupuncture had a better therapeutic effect compared with combined rehabilitation training (SMD = -0.38, 95% CI [-0.6, -0.16], Z = 3.41, and *P* < .05) (Fig 9B).

A

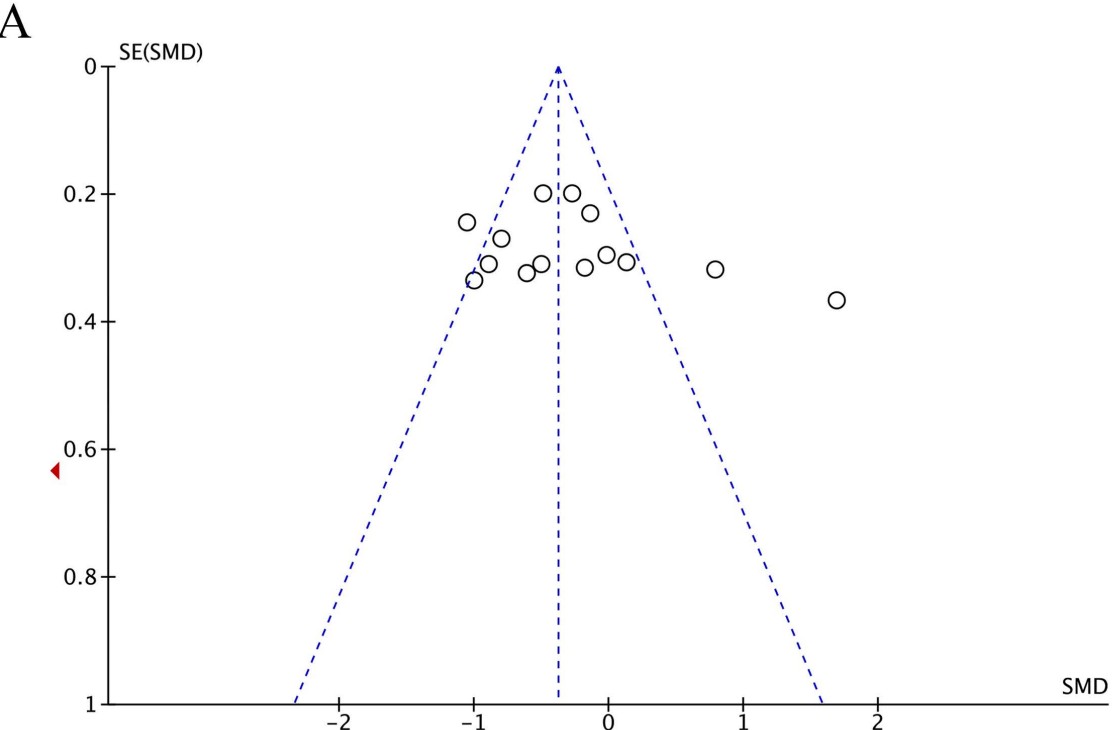

B

```
Begg's Test

    adj. Kendall's Score (P-Q) =        -1
            Std. Dev. of Score =     20.21
            Number of Studies =        15
                           z  =     -0.05
                  Pr > |z|  =      0.961
                           z  =      0.00 (continuity corrected)
                  Pr > |z|  =      1.000 (continuity corrected)

Egger's test
```

| Std_Eff | Coefficient | Std. err. | t | P>\|t\| | [95% conf. interval] | |
|---|---|---|---|---|---|---|
| slope | .4674757 | .9366564 | 0.50 | 0.626 | -1.556047 | 2.490999 |
| bias | -3.108913 | 3.332482 | -0.93 | 0.368 | -10.3083 | 4.090477 |

**Fig 5. FMA-L scale publication bias test. (A)** The funnel plot of FMA-L. **(B)** The Egger test result of FMA-L.

A

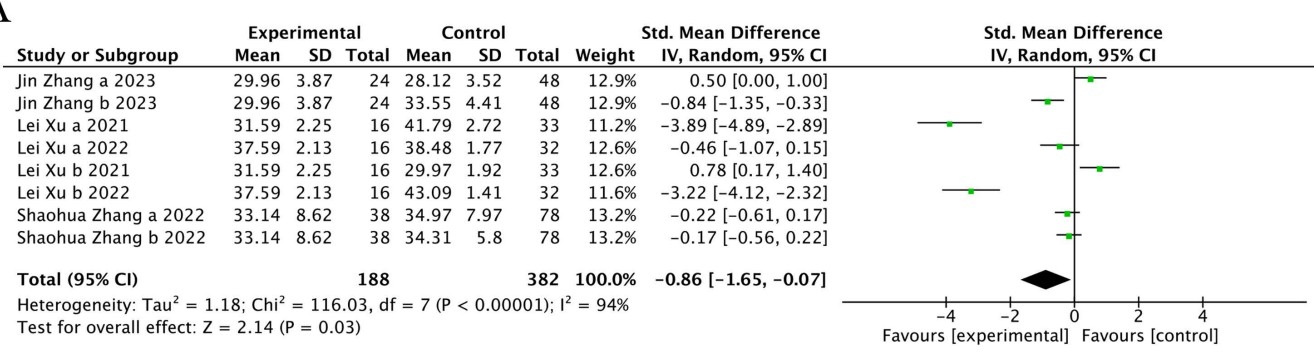

B

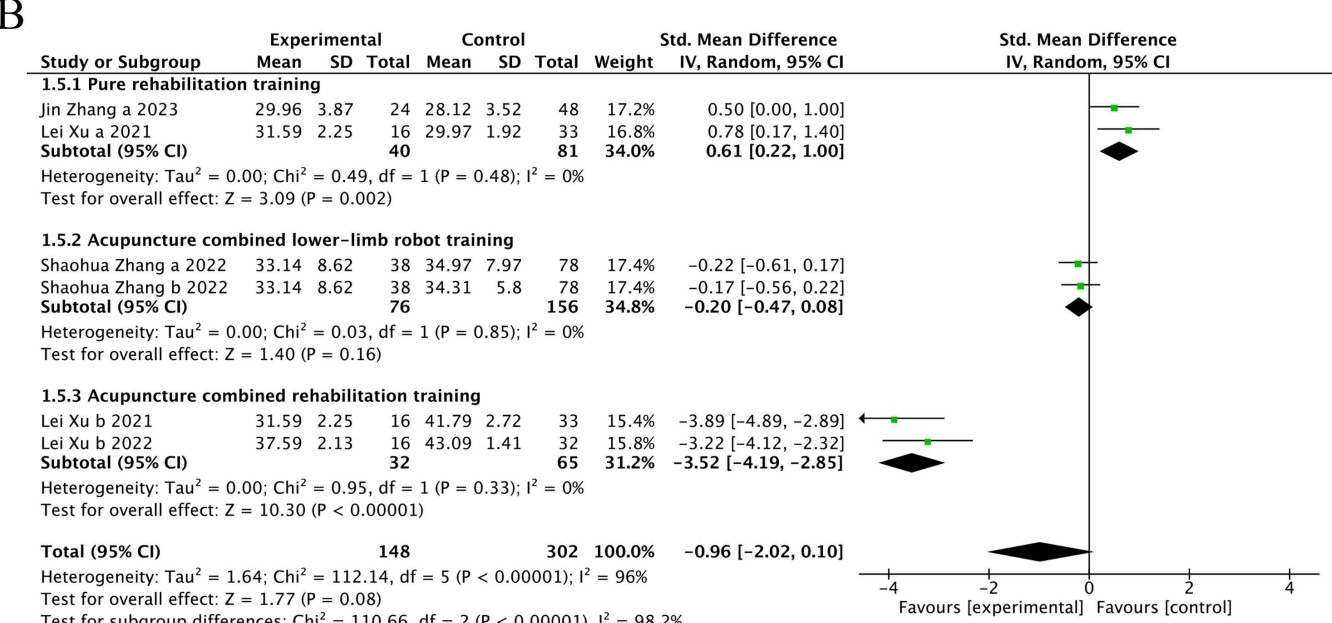

**Fig 6. The forest plot of BBS. (A)** Meta-analysis results containing BBS for all studies. **(B)** Results of subgroup analyses of BBS.

Sensitivity analysis revealed the results lacked robustness (see Fig 10). When examining the possibility of publication bias, both the funnel plot (Fig 11A) and Egger's test (P=0.018<0.05, Fig 11B) indicated no evidence of such bias.

**3.4.4. FAC score.** Two trials [28,32] (16.67%) involving 312 participants reported FAC as the primary outcome. The experimental group did not show significant improvement in FAC compared with the control group (SMD=-0.74, 95%CI[-2.33, 0.84], I2=97%, P=.36, random-effects mode) (Fig 12A). There was significant heterogeneity. Subgroup analysis was used to explore the source of heterogeneity. Subgroup analyses were performed according to the different intervention modalities in the control group. The results showed that acupuncture intervention was not statistically significant compared with pure rehabilitation (SMD=0.62, 95% CI [-0.69, 1.92], Z=0.92, P=.35), Similarly, when compared with combined rehabilitation training, acupuncture showed no statistically significant therapeutic effect (SMD=-2.13, 95% CI [-4.58, 0.31], Z=1.71, and P=.09) (Fig 12B).

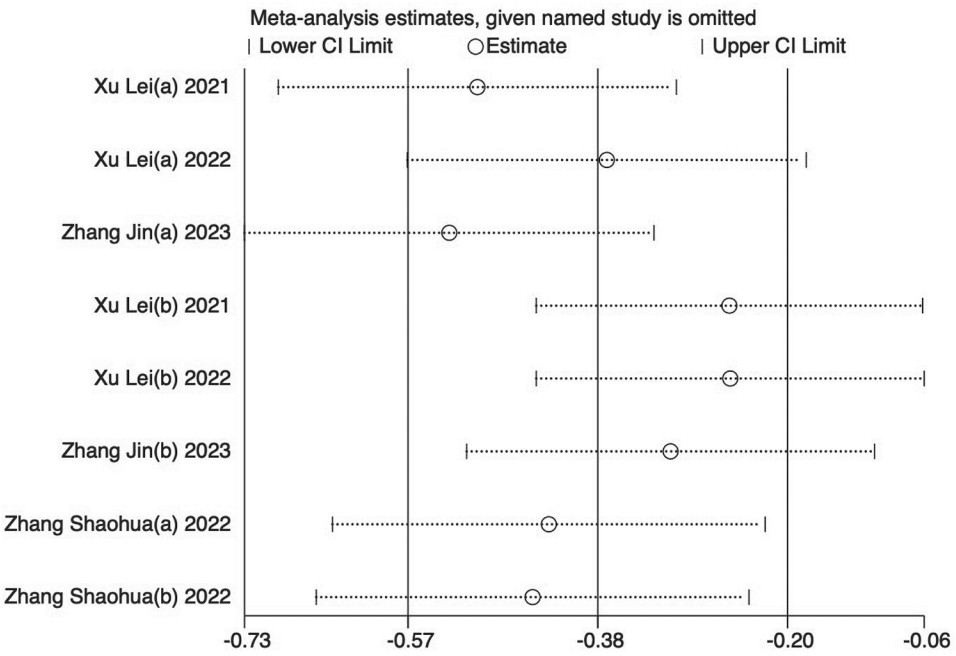

**Fig 7. Sensitivity analysis results for BBS.**

## 4. Discussion

To address critical questions regarding post-stroke motor dysfunction and its treatment approaches, this meta-analysis of randomized clinical trials synthesized data on the therapeutic effects of acupuncture, conventional rehabilitation training alone, and combined rehabilitation approaches in improving post-stroke motor impairment. Our findings suggest that while acupuncture demonstrated positive effects on motor function and balance control, its efficacy in enhancing walking ability and activities of daily living was limited.

Our findings align with the functional magnetic resonance imaging study by Wang et al., which provided neuroimaging evidence demonstrating that acupuncture significantly modulates dynamic functional network connectivity in stroke patients' brains [35]. However, some discrepancies exist between our findings and those of Zhu et al., which may be attributed to differences in assessment timepoints, sensitivity of evaluation tools, and heterogeneity in treatment protocols [20]. While a recent network meta-analysis systematically evaluated the effects of various acupuncture methods, it lacked long-term follow-up data [36]. In contrast, our study enhanced the reliability of results through more stringent inclusion criteria and standardized quality assessment tools.

Our findings revealed that acupuncture treatment significantly improved lower limb motor function and balance control, suggesting that acupuncture may be particularly effective in postural control rehabilitation. However, we observed no significant improvements in activities of daily living and walking ability across different intervention approaches. This suggests that selecting tailored interventions based on individual patient conditions may yield optimal therapeutic outcomes.

Subsequent subgroup analyses revealed that acupuncture demonstrated superior efficacy compared to conventional rehabilitation training alone, with even greater improvements observed in combined rehabilitation interventions. These differential outcomes suggest that acupuncture may enhance rehabilitation training effects through specific mechanisms rather than simple additive effects. Through heterogeneity analysis, we identified moderate heterogeneity among studies

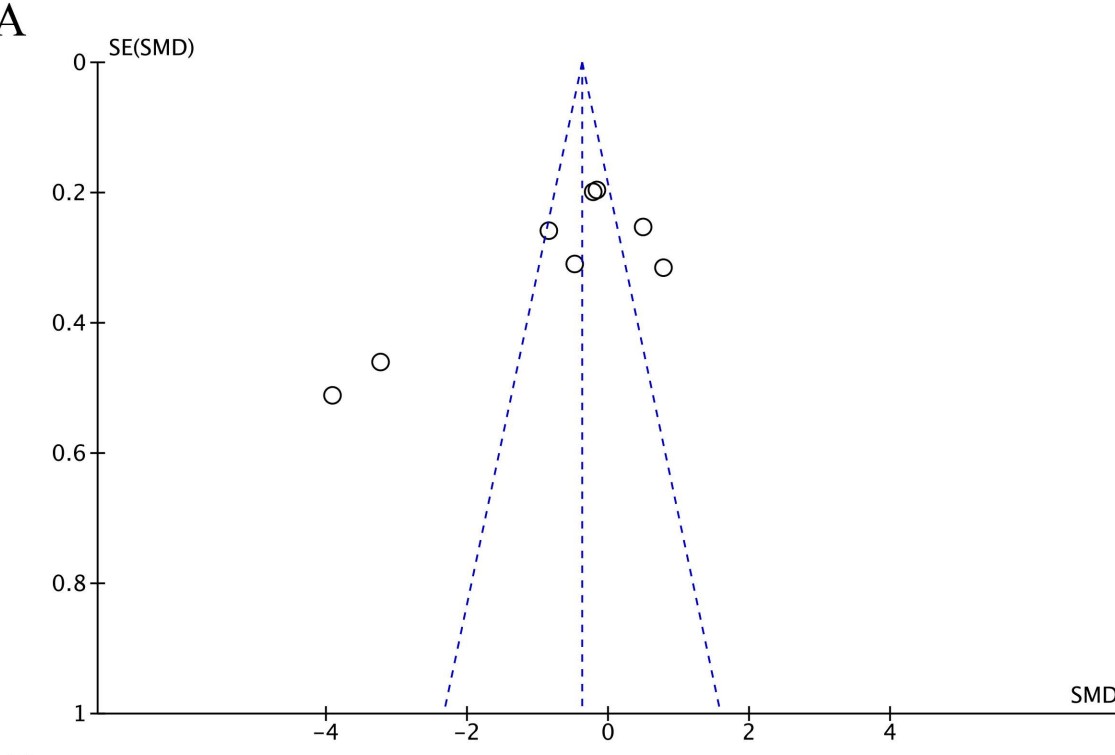

A

B

Begg's Test

```
   adj. Kendall's Score (P-Q) =       -12
           Std. Dev. of Score =      8.08
           Number of Studies =         8
                          z  =     -1.48
                  Pr > |z|  =     0.138
                          z  =      1.36 (continuity corrected)
                  Pr > |z|  =     0.174 (continuity corrected)
```

Egger's test

| Std_Eff | Coefficient | Std. err. | t | P>\|t\| | [95% conf. interval] | |
|---|---|---|---|---|---|---|
| slope | 1.980677 | 1.061621 | 1.87 | 0.111 | -.617016 | 4.578369 |
| bias | -9.178505 | 3.94138 | -2.33 | 0.059 | -18.82271 | .465704 |

**Fig 8. BBS scale publication bias test. (A)** The funnel plot of BBS. **(B)** The Egger test result of BBS.

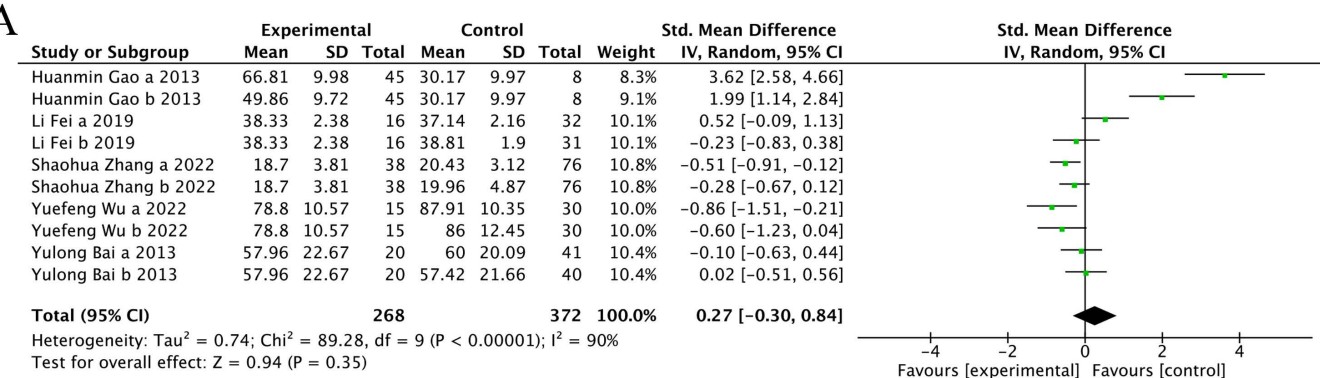

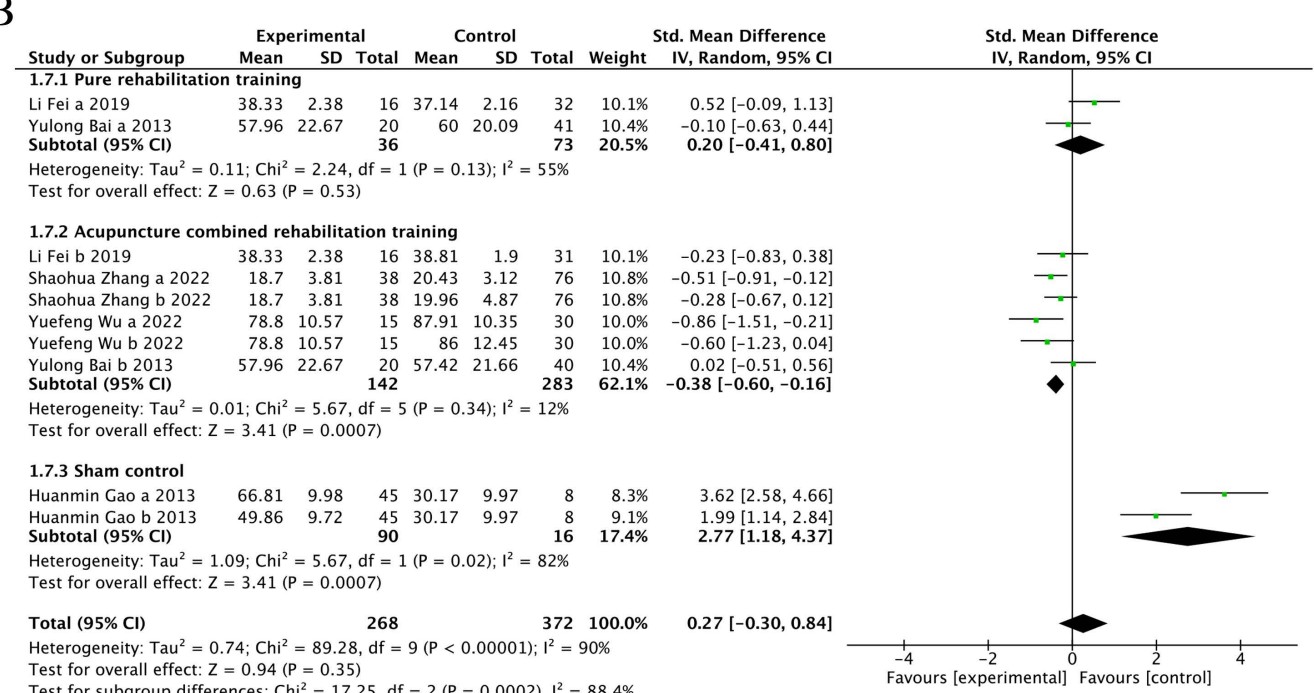

**Fig 9. The forest plot of MBI. (A)** Meta-analysis results containing MBI for all studies. **(B)** Results of subgroup analyses of MBI.

(I²=46%, P=0.03), primarily attributed to variations in acupuncture protocols, patient baseline characteristics, and rehabilitation training regimens, rather than common factors such as treatment duration, disease course, or baseline levels. Results remained stable after applying random-effects models and sensitivity analyses, with funnel plots and Egger's test showing no significant publication bias.

Our findings did not demonstrate improvements in activities of daily living and walking ability with acupuncture, conventional rehabilitation training alone, or combined interventions. This observation may be attributed to the limited number of studies included in the MBI and FAC analyses, with only four RCTs incorporated in the FAC analysis. However, two recent meta-analyses have indicated that acupuncture combined with rehabilitation training is associated with improvements in daily functional activities and walking function, suggesting the need for more high-quality RCTs to further validate these findings [37,38].

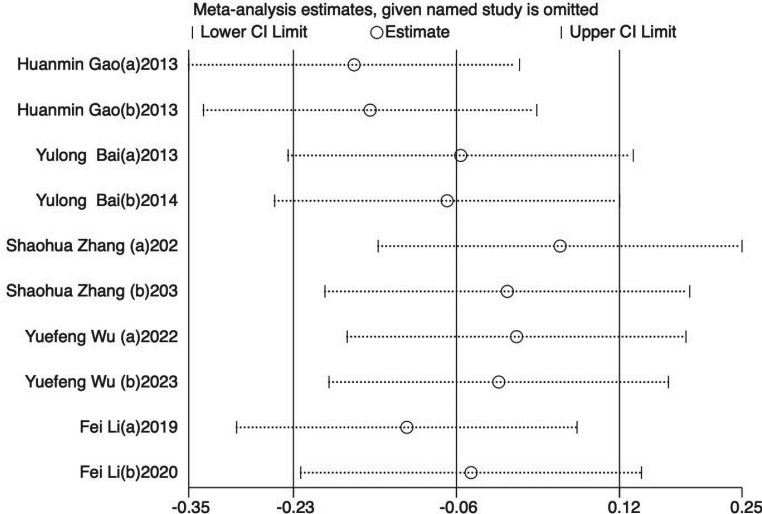

**Fig 10. Sensitivity analysis results for MBI.**

Based on existing fundamental research evidence, the mechanisms by which acupuncture improves LLMD likely involve multiple levels. Regarding neuroplasticity, acupuncture upregulates the expression of neurotrophic factors such as BDNF and NGF, promotes dendritic remodeling and synapse formation, and enhances functional reconstruction of the corticospinal tract [39]. In terms of local microcirculation, acupuncture increases local blood perfusion, improves capillary density, and promotes tissue oxygenation [40]. For neuromuscular function, acupuncture optimizes motor patterns and facilitates functional reconstruction of synergistic muscle groups. When combined with rehabilitation training, acupuncture creates favorable conditions for rehabilitation by modulating neural excitability, while rehabilitation training further reinforces the neuroplastic changes induced by acupuncture. These findings have significant implications for clinical practice. The notable improvements in FMA-L and BBS scores provide evidence-based support for incorporating acupuncture into post-stroke rehabilitation systems. Subgroup analysis results suggest that combined acupuncture and rehabilitation training may be the optimal treatment choice, particularly for patients in acute and subacute stroke phases. For patients with severe motor impairment, acupuncture can serve as a safe and effective early intervention strategy. However, more comprehensive rehabilitation strategies may be necessary to improve activities of daily living.

Several limitations of this study need to be evaluated in future research and should be considered when interpreting the results. First, the high heterogeneity in the data may limit the generalizability of our findings. This may be attributed to the lack of standardization in specific acupuncture types, although acupuncture was the primary intervention in the experimental groups, and variations in acupuncture protocols across studies made it difficult to determine optimal treatment parameters. Second, the variability in control group interventions complicated direct comparisons between studies. Additionally, the number of available trials was limited for certain outcomes, such as improvements in activities of daily living and walking function. Finally, only one study compared acupuncture with placebo control, and most studies had relatively short follow-up periods, which limited our understanding of acupuncture-specific effects and long-term outcomes.

Based on these findings, future research should focus on conducting well-designed, large-scale multicenter randomized controlled trials with emphasis on developing standardized treatment protocols. Long-term follow-up studies are needed to evaluate the sustained effects of acupuncture interventions. Additionally, studies should explore optimal combinations and timing of acupuncture with rehabilitation training, and employ more objective assessment tools. These

A

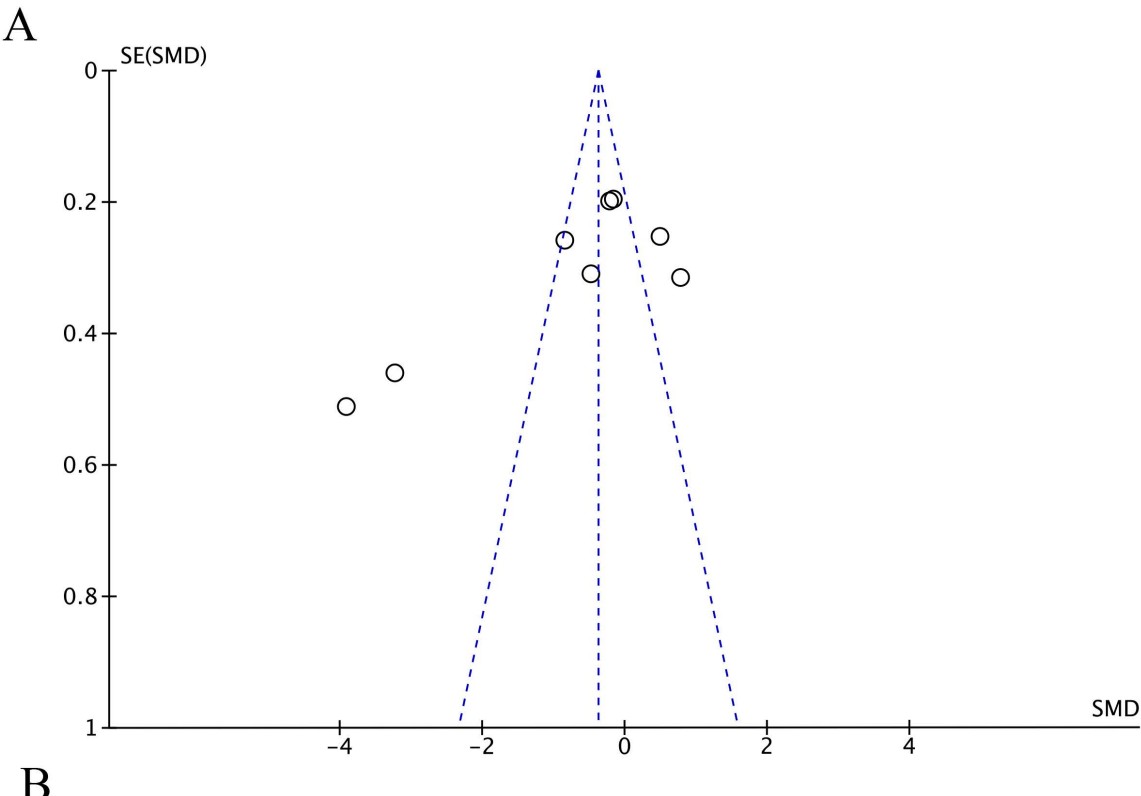

B

```
Begg's Test

    adj. Kendall's Score (P-Q) =        15
              Std. Dev. of Score =     11.18
             Number of Studies =        10
                           z  =       1.34
                  Pr > |z|  =       0.180
                           z  =       1.25 (continuity corrected)
                  Pr > |z|  =       0.210 (continuity corrected)

Egger's test
```

| Std_Eff | Coefficient | Std. err. | t | P>\|t\| | [95% conf. interval] | |
|---|---|---|---|---|---|---|
| slope | −2.302773 | .7847941 | −2.93 | 0.019 | −4.112512 | −.4930347 |
| bias | 8.229413 | 2.774479 | 2.97 | 0.018 | 1.831454 | 14.62737 |

**Fig 11. MBI scale publication bias test. (A)** The funnel plot of MBI. **(B)** The Egger test result of MBI.

A

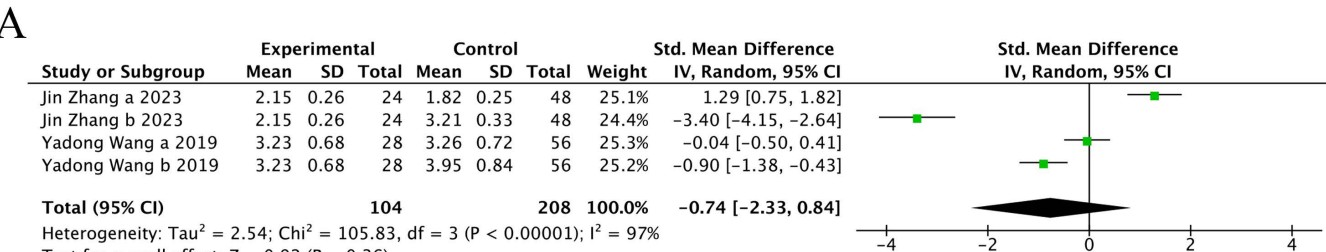

B

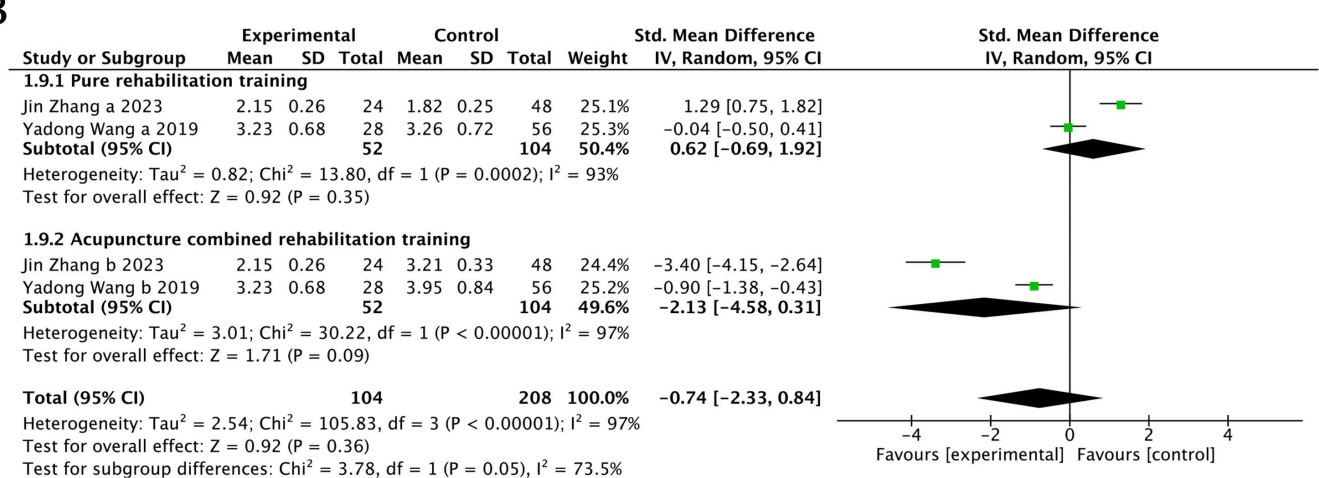

**Fig 12. The forest plot of FAC. (A)** Meta-analysis results containing FAC for all studies. **(B)** Results of subgroup analyses of FAC.

efforts will help establish stronger evidence-based support to further guide the clinical application of acupuncture in stroke rehabilitation.

## 5. Conclution

This study proved that the efficacy of acupuncture for LLMD is inconclusive. But, the results of this study suggest that acupuncture combined rehabilitation training has clear superiority for LLMD compared with pure acupuncture or pure rehabilitation training, for provide evidence-based support for further research and promote the effects of acupuncture treatment on LLMD in poststroke patients. Influenced by the quality and quantity of the included literature, future studies should focus on better-designed and more comprehensive outcome indicators as well as multi-center and high-quality RCT.

## Author contributions

**Conceptualization:** Bin Shao.

**Data curation:** Debiao Yu.

**Funding acquisition:** Bin Shao.

**Methodology:** Xiaoting Chen.

**Supervision:** Fuchun Wu.

**Validation:** Yaoyu Lin.

**Writing – original draft:** Xing Jin.

**Writing – review & editing:** Peng Chen.

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
