## [Decision Letter · Decision Letter 0]

27 Nov 2024

PONE-D-24-45171Efficacy of Acupuncture on Lower Limb Motor Dysfunction Following Stroke: A Systematic Review and Meta-Analysis of Randomized Controlled Trials.PLOS ONE

Dear Dr. Shao,

Thank you for submitting your manuscript to PLOS ONE. After careful consideration, we feel that it has merit but does not fully meet PLOS ONE’s publication criteria as it currently stands. Therefore, we invite you to submit a revised version of the manuscript that addresses the points raised during the review process.

We look forward to receiving your revised manuscript.

Kind regards,

Sangharsha Thapa

Academic Editor

PLOS ONE

Journal Requirements:

3. Thank you for stating the following financial disclosure: “Fund Project of Fujian Provincial Natural Science Foundation in 2022 (No. 2022J011019) and the 2022 Fujian Provincial Health and Health Youth Backbone Training Project (No. 2022GA009)”.

4. We note that your Data Availability Statement is currently as follows: “All relevant data are within the manuscript and in Supporting Information files.”

Please confirm at this time whether or not your submission contains all raw data required to replicate the results of your study. Authors must share the “minimal data set” for their submission. PLOS defines the minimal data set to consist of the data required to replicate all study findings reported in the article, as well as related metadata and methods (https://journals.plos.org/plosone/s/data-availability#loc-minimal-data-set-definition). For example, authors should submit the following data: - The values behind the means, standard deviations and other measures reported; - The values used to build graphs; - The points extracted from images for analysis. Authors do not need to submit their entire data set if only a portion of the data was used in the reported study. If your submission does not contain these data, please either upload them as Supporting Information files or deposit them to a stable, public repository and provide us with the relevant URLs, DOIs, or accession numbers. For a list of recommended repositories, please see https://journals.plos.org/plosone/s/recommended-repositories. If there are ethical or legal restrictions on sharing a de-identified data set, please explain them in detail (e.g., data contain potentially sensitive information, data are owned by a third-party organization, etc.) and who has imposed them (e.g., an ethics committee). Please also provide contact information for a data access committee, ethics committee, or other institutional body to which data requests may be sent. If data are owned by a third party, please indicate how others may request data access.

6. PLOS requires an ORCID iD for the corresponding author in Editorial Manager on papers submitted after December 6th, 2016. Please ensure that you have an ORCID iD and that it is validated in Editorial Manager. To do this, go to ‘Update my Information’ (in the upper left-hand corner of the main menu), and click on the Fetch/Validate link next to the ORCID field. This will take you to the ORCID site and allow you to create a new iD or authenticate a pre-existing iD in Editorial Manager.

7. Please include your tables as part of your main manuscript and remove the individual files. Please note that supplementary tables (should remain/ be uploaded) as separate "supporting information" files

9. As required by our policy on Data Availability, please ensure your manuscript or supplementary information includes the following: A numbered table of all studies identified in the literature search, including those that were excluded from the analyses. For every excluded study, the table should list the reason(s) for exclusion. If any of the included studies are unpublished, include a link (URL) to the primary source or detailed information about how the content can be accessed. A table of all data extracted from the primary research sources for the systematic review and/or meta-analysis. The table must include the following information for each study: Name of data extractors and date of data extraction Confirmation that the study was eligible to be included in the review. All data extracted from each study for the reported systematic review and/or meta-analysis that would be needed to replicate your analyses. If data or supporting information were obtained from another source (e.g. correspondence with the author of the original research article), please provide the source of data and dates on which the data/information were obtained by your research group. If applicable for your analysis, a table showing the completed risk of bias and quality/certainty assessments for each study or outcome. Please ensure this is provided for each domain or parameter assessed. For example, if you used the Cochrane risk-of-bias tool for randomized trials, provide answers to each of the signalling questions for each study. If you used GRADE to assess certainty of evidence, provide judgements about each of the quality of evidence factor. This should be provided for each outcome. An explanation of how missing data were handled. This information can be included in the main text, supplementary information, or relevant data repository. Please note that providing these underlying data is a requirement for publication in this journal, and if these data are not provided your manuscript might be rejected.

Additional Editor Comments:

Hello ,

The reviewer's has commented the paper and requested for these minor revision . The paper would be considerable with these comment's get addressed

Reviewers' comments:

Reviewer's Responses to Questions

**Comments to the Author**

1. Is the manuscript technically sound, and do the data support the conclusions?

Reviewer #1: Yes

Reviewer #2: Yes

2. Has the statistical analysis been performed appropriately and rigorously? 

Reviewer #1: Yes

Reviewer #2: Yes

3. Have the authors made all data underlying the findings in their manuscript fully available?

Reviewer #1: Yes

Reviewer #2: Yes

4. Is the manuscript presented in an intelligible fashion and written in standard English?

Reviewer #1: Yes

Reviewer #2: No

5. Review Comments to the Author

Reviewer #1: 1. General Evaluation

The manuscript titled "Efficacy of Acupuncture on Lower Limb Motor Dysfunction Following Stroke: A Systematic Review and Meta-Analysis of Randomized Controlled Trials" addresses the effectiveness of acupuncture for lower limb motor dysfunction in post-stroke patients. The study is relevant, considering the growing impact of stroke on public health and the challenges in motor rehabilitation. The analysis involves a meta-analysis of randomized controlled trials and utilizes recognized assessment parameters, such as the Fugl–Meyer Assessment (FMA-L), Berg Balance Scale (BBS), and Modified Barthel Index (MBI). The manuscript is well-structured and organized, but it presents important methodological limitations.

2. Objective and Rationale

The objective of evaluating the efficacy of acupuncture in improving lower limb motor function post-stroke is clearly established and well justified, based on literature data on the increasing prevalence of stroke and the role of acupuncture in traditional Chinese medicine. However, it would be advisable for the authors to describe in more detail the rationale behind the selection of specific outcomes assessed, as well as to include an introductory section on the possible mechanisms of action of acupuncture.

3. Methodology

The study adopts a robust methodology, searching eight databases for the selection of randomized controlled trials and conducting a rigorous data analysis with specialized software.

Inclusion and Exclusion Criteria: The inclusion criteria are well defined, covering only randomized controlled trials published in English and Chinese. However, excluding studies with combined acupuncture interventions may limit the generalizability of the findings, given the diversity of acupuncture techniques used.

Risk of Bias Analysis: The authors used Cochrane methods for risk of bias assessment, but only one included study applied allocation concealment. This suggests a high risk of bias and limitations in interpreting the results.

Statistical Methods: The choice of a random effects model is appropriate, given the heterogeneity among studies. The sensitivity analysis and publication bias tests contribute to the robustness of the conclusions. However, high heterogeneity in some analyses requires further exploration, which could include additional subgroup analyses based on specific study characteristics.

4. Results

The meta-analysis results suggest that acupuncture may benefit lower limb motor function and balance, although the benefits for walking ability and daily activities are limited.

Heterogeneity and Subgroup Analysis: Subgroup analysis is an appropriate approach to explore heterogeneity, but the findings lack robustness, raising questions about the consistency of the results.

Outcome Reporting: Outcomes are well presented, but the interpretation of effect sizes is limited by insufficient description of the magnitude of the observed clinical benefits.

5. Discussion and Conclusions

The discussion compares the findings with the existing literature and mentions the limitations, including risk of bias and methodological variability of the included studies. However, the authors could further explore limitations related to the diversity of acupuncture interventions and the absence of blinding.

Limitations: Limitations are mentioned in general terms, but a deeper analysis of the impact of selection bias and lack of blinding in the included studies would improve the transparency and reliability of the conclusions.

Suggestions for Future Research: The recommendation for future studies is appropriate, but the authors could provide more details on study designs that might mitigate the limitations identified in this study

Reviewer #2: The authors present a systematic review with meta-analysis of the effects of acupuncture alone and/or combined with rehabilitation on lower extremity motor dysfunction. A key conclusion was that acupuncture combined with rehabilitation was superior to acupuncture alone or rehabilitation alone.

The paper is substantially limited by lacking a strong theoretical framework for how acupuncture works at a physiological level. It is well known that acupuncture is popular in China, but that does little to help explain findings such as greater improvement in Berg Balance scale with acupuncture alone than rehabilitation alone (and other findings). I would revise the introduction and discussion to provide a stronger physiologically based theoretical framework.

Minor concerns

There are several grammatical errors. Please resolve. If necessary, please have the manuscript reviewed by a native English speaker.

There are a large number of abbreviations that detract from overall readability. Please revise.

Including figure captions in-text without associated figures made reading the manuscript more complicated. Unless this was a requirement of PLoS ONE, I would relocate those for any subsequent revisions.

The X-axes for the forest plots are not clear. Likewise, associated text in the results section are difficult to understand. Please re-write to improve clarity.

6. PLOS authors have the option to publish the peer review history of their article (what does this mean? ). If published, this will include your full peer review and any attached files.

**Do you want your identity to be public for this peer review?** For information about this choice, including consent withdrawal, please see our Privacy Policy .

Reviewer #1: No

Reviewer #2: No

---

## [Author Response · Author response to Decision Letter 0]

21 Jan 2025

Response to Comment 1: We appreciate your attention to the formatting requirements. We have thoroughly revised our manuscript to ensure full compliance with PLOS ONE's style requirements. Specifically:

The manuscript has been reformatted according to the PLOS ONE formatting templates

All sections now follow the required style guidelines. File naming conventions have been adjusted to meet the journal's standards. The title page, authors' affiliations, and main body formatting have been updated to match the provided template examples. The revised manuscript now fully adheres to the journal's formatting requirements as detailed in the provided template documents.

Response to Comment 2: We thank you for bringing this to our attention. We have carefully reviewed the manuscript and removed all funding-related information from the main text. The funding information is now only included in the Funding Statement section of the online submission form, in accordance with PLOS ONE's requirements.

Response to Comment 3: We confirm that the funding information is correct: "Fund Project of Fujian Provincial Natural Science Foundation in 2022 (No. 2022J011019) and the 2022 Fujian Provincial Health and Health Youth Backbone Training Project (No. 2022GA009)".

Regarding the role of funders, we confirm that: The funders had no role in study design, data collection and analysis, decision to publish, or preparation of the manuscript."

Response to Comment 4: Thank you for your inquiry regarding the Data Availability Statement. As this is a meta-analysis, we confirm that all data used in our analysis are drawn from previously published studies, and we have provided comprehensive information in the following forms:

All data extracted from the included studies are presented in the manuscript and Supporting Information files. Complete extraction forms showing all data points used in our meta-analysis. The full search strategy and selection criteria. All statistical analysis details, including the values used to generate forest plots and other meta-analytical figures. A detailed PRISMA flow diagram showing the study selection process. The complete reference list of all included studies.

The data presented in our manuscript allows for full transparency and replication of our meta-analysis findings. All original data can be accessed through the published articles cited in our reference list.

Response to Comment 5: We fully understand and respect PLOS ONE's open data policy. As this is a meta-analysis study, we will make available all the extracted data, including:

The complete data extraction sheets. The statistical analysis files. The search strategy and results. The quality assessment forms for included studies. The detailed calculations for effect sizes.

We will deposit these materials in Open Science Framework (OSF) immediately upon acceptance of our manuscript. All these data are derived from published studies and sharing them will not raise any ethical concerns. This comprehensive data sharing will enable other researchers to verify our findings and potentially conduct additional analyses, supporting research transparency and reproducibility.

Response to Comment 6: Thank you for bringing this to our attention. We confirm that the corresponding author has already registered and validated their ORCID iD in the Editorial Manager system following the required procedure. The ORCID iD has been successfully linked to their profile, ensuring compliance with PLOS requirements for manuscript submission.

Response to Comment 7: We have now incorporated all main tables into the manuscript text and removed the individual table files as requested. All supplementary tables have been properly uploaded as separate supporting information files. We have ensured that the tables in the main manuscript are properly formatted and numbered sequentially, while maintaining all supplementary tables as separate supporting information files.

Thank you for this reminder about Supporting Information formatting. We have now added detailed captions for all Supporting Information files at the end of our manuscript. All in-text citations of Supporting Information have been carefully reviewed and updated to ensure they match with the corresponding captions. The formatting follows PLOS ONE's Supporting Information guidelines. Specifically:

Response to Comment 7: All Supporting Information is now labeled as 'S1', 'S2', etc., followed by the file type (Table, Figure, etc.). Complete captions have been added at the end of the manuscript. All in-text citations have been updated to match these labels. The format complies with the journal's Supporting Information guidelines.

Response to Comment 9: We have carefully addressed all requirements with the following specific modifications:

1. Literature Search Documentation: We have provided a complete literature screening table (Table S1) in the supplementary materials. As shown in Figure 1, we have documented:The number of articles from each database (total 2,718, including 512 from PubMed, 687 from Embase, etc.). Excluded duplicates (n=775) and records excluded for other reasons. Step-by-step screening process with specific exclusion reasons. Final inclusion of 12 studies

2. Data Extraction Documentation: We have provided a detailed data extraction table (Table S2), as shown in Figure 2, including: Basic information for each study (authors, year). Sample size and gender ratio. Course of disease. Age information. Interventions. Outcome measures

3. Risk of Bias Assessment: As shown in Figure 3, we have provided comprehensive risk of bias assessment results: Evaluated key domains including random sequence generation, allocation concealment, blinding, etc. Used red, yellow, and green to indicate high, unclear, and low risk of bias. Provided specific ratings for each domain of every study. Included both overall risk of bias graph and detailed assessment for individual studies

4. Missing Data Handling: We have explicitly described our approach to handling missing data in the Methods section, ensuring transparency in our analysis process.

Response to Comment 10: We have carefully reviewed all references to ensure their completeness and accuracy:

1. We have verified each reference for: Accurate article titles; Author names and order; Journal names; Publication years; Volume and page numbers; DOI numbers

2. After thorough examination: All cited references are valid and have not been retracted. In-text citations correctly correspond to the reference list. Reference format complies with journal requirements

Response to Reviewer #1: Thank you for your valuable feedback. We have made the following revisions based on your suggestions:

In the Methods section, we have clarified the rationale and limitations of our chosen assessment tools (FMA-L, BBS, and MBI).

In Section 4 of the Discussion, we have expanded on the mechanisms by which acupuncture improves post-stroke motor function, including neuroplasticity, local microcirculation, and neuromuscular function.

In the Statistical Analysis section, we have incorporated subgroup and sensitivity analyses to explore heterogeneity between studies and evaluate the robustness of our findings.

We have acknowledged the limitations in assessing walking ability and activities of daily living, and provided recommendations for future research.

Response to Reviewer #2: Thank you for your insightful comments on the theoretical framework of acupuncture's physiological mechanisms. We have:

Strengthened the discussion of acupuncture's physiological mechanisms

Detailed why combined rehabilitation might be more effective than monotherapy

Acknowledged the cultural context of acupuncture while maintaining focus on evidence-based outcomes

Expanded the discussion of Berg Balance Scale improvements within the context of overall rehabilitation outcomes

We believe these revisions have enhanced the scientific rigor of our study. We welcome any additional suggestions for further improvements.

---

## [Decision Letter · Decision Letter 1]

19 Feb 2025

PONE-D-24-45171R1Efficacy of Acupuncture on Lower Limb Motor Dysfunction Following Stroke: A Systematic Review and Meta-Analysis of Randomized Controlled Trials.PLOS ONE

Dear Dr. Shao,

Thank you for submitting your manuscript to PLOS ONE. After careful consideration, we feel that it has merit but does not fully meet PLOS ONE’s publication criteria as it currently stands. Therefore, we invite you to submit a revised version of the manuscript that addresses the points raised during the review process before accepting the manuscript for publication

We look forward to receiving your revised manuscript.

Kind regards,

Sangharsha Thapa

Academic Editor

PLOS ONE

Journal Requirements:

Additional Editor Comments:

Reviewer #2: I appreciate the authors efforts in addressing my previous comments. I believe that largely my concerns have been addressed. However, I do think that some of the changes to setting up accupuncture from a physiological perspective need to be present in the Introduction. Please make changes to the Introduction itself to better introduce this topic.

The changes to the discussion are under cited. Please add in relevant citations throughout. 

Reviewers' comments:

Reviewer's Responses to Questions

**Comments to the Author**

1. If the authors have adequately addressed your comments raised in a previous round of review and you feel that this manuscript is now acceptable for publication, you may indicate that here to bypass the “Comments to the Author” section, enter your conflict of interest statement in the “Confidential to Editor” section, and submit your "Accept" recommendation.

Reviewer #1: All comments have been addressed

Reviewer #2: (No Response)

2. Is the manuscript technically sound, and do the data support the conclusions?

Reviewer #1: Yes

Reviewer #2: Yes

3. Has the statistical analysis been performed appropriately and rigorously? 

Reviewer #1: Yes

Reviewer #2: Yes

4. Have the authors made all data underlying the findings in their manuscript fully available?

Reviewer #1: Yes

Reviewer #2: Yes

5. Is the manuscript presented in an intelligible fashion and written in standard English?

Reviewer #1: Yes

Reviewer #2: Yes

6. Review Comments to the Author

Reviewer #1: The revision significantly improved the clarity and objectivity of the manuscript. The responses to the reviewers indicate an effort to improve the formatting, correct inaccuracies and better structure the sections. There was also greater detail in the methods and inclusion of essential information, which strengthens the quality of the work.

Reviewer #2: I appreciate the authors efforts in addressing my previous comments. I believe that largely my concerns have been addressed. However, I do think that some of the changes to setting up accupuncture from a physiological perspective need to be present in the Introduction. Please make changes to the Introduction itself to better introduce this topic.

The changes to the discussion are under cited. Please add in relevant citations throughout.

7. PLOS authors have the option to publish the peer review history of their article (what does this mean? ). If published, this will include your full peer review and any attached files.

**Do you want your identity to be public for this peer review?** For information about this choice, including consent withdrawal, please see our Privacy Policy .

Reviewer #1: No

Reviewer #2: No

---

## [Author Response · Author response to Decision Letter 1]

10 Mar 2025

Dear Editor,

We thank you and the reviewers for your careful evaluation of our manuscript. We have addressed all the comments and made the necessary revisions to improve our paper. Below, we provide point-by-point responses to each of the concerns raised.

Response to Editor's Comments:

We have carefully reviewed our reference list and confirmed that all citations are complete, accurate, and up-to-date. No retracted articles were cited in our manuscript.

Response to Reviewer #2:

We thank the reviewers for their positive feedback on our previous revisions. As suggested, we have expanded the introductory section to better explain the physiological mechanisms of acupuncture. Specifically, we have added: 1. an overview of the known neurophysiological pathways that are activated during acupuncture stimulation, including the role of the peripheral and central nervous systems. 2. A discussion of the mechanisms by which acupuncture improves motor dysfunction after stroke.

We believe these revisions have enhanced the scientific rigor of our study. We welcome any additional suggestions for further improvements.

---

## [Editor Report · Decision Letter 2]

25 Mar 2025

Efficacy of Acupuncture on Lower Limb Motor Dysfunction Following Stroke: A Systematic Review and Meta-Analysis of Randomized Controlled Trials.

PONE-D-24-45171R2

Dear Dr. Shao,

We’re pleased to inform you that your manuscript has been judged scientifically suitable for publication and will be formally accepted for publication once it meets all outstanding technical requirements.

Kind regards,

Sangharsha Thapa

Academic Editor

PLOS ONE
---

## [Editor Report · Acceptance letter]

PONE-D-24-45171R2

PLOS ONE

Dear Dr. Shao,

I'm pleased to inform you that your manuscript has been deemed suitable for publication in PLOS ONE. Congratulations! Your manuscript is now being handed over to our production team.

Kind regards,

on behalf of

Dr. Sangharsha Thapa

Academic Editor

PLOS ONE